# Tuning the Surface Characteristics and Mechanical Properties of Y_2_O_3_ Coatings on a Graphene Matrix via Laser Micro Melting

**DOI:** 10.3390/ma15155443

**Published:** 2022-08-08

**Authors:** Hao Liu, Ping-hu Chen, Yong Chen, Wen-xing Wu, Sheng Li, Chang-jun Qiu

**Affiliations:** 1School of Resource Environment and Safety Engineering, College of Mechanical Engineering, School of Nuclear Science and Technology, University of South China, Hengyang 421001, China; 2College of Mechatronics & Control Engineering, Shenzhen University, Shenzhen 518060, China

**Keywords:** plasma spray, critical load, laser micro-melting, mechanical properties

## Abstract

The effects of laser parameters on the microstructure and properties of plasma-sprayed yttrium oxide coating on the graphite matrix were investigated. Tensile strength, porosity, roughness, and scratch meter tests were carried out to evaluate the critical load and mechanical properties of the coating after spraying and laser micro-melting. When the porosity and surface roughness of the coating are minimum, the critical load of the coating is 7.85 N higher than that of the spraying surface. After laser micromelting, the crystal phase of Y_2_O_3_ coating surface does not change, the crystallinity is improved, and fine grain strengthening occurs. When the laser power density is 75 W/mm^2^, the scanning speed is 30 mm/s, and the defocusing distance is 40 mm, the film base bonding performance and wear resistance of the material reach the maximum value. The failure of Y_2_O_3_ coating is mainly due to the degradation of mechanical properties such as film base bonding strength, surface porosity, and surface roughness, which leads to the local collapse of the material. The coating after laser micro-melting only presents particle disintegration at the end of the scratch area.

## 1. Introduction

High density graphite (HDG) crucible is recommended for u-Zr smelting in high-temperature chemical reprocessing because of its improved performance and durability to withstand multiple melts, thereby reducing the generation of active solid waste [1]. However, it will undergo rapid oxidation and erosion above 723 K [2], thus limiting its application in extreme temperature environments [3]. This requires providing the HDG with a ceramic coating to protect it from oxidation at high temperatures and extend its service life [4]. Yttrium oxide (Y_2_O_3_) is an ultra-high temperature ceramic (UHTC) coating material used in aerospace and nuclear material forming systems [5,6]. It can be used in high temperature chemical post-processing in smelting processes to improve the performance and durability of the smelting process [7,8]. Molybdenum (Mo), as a transition layer, reacts with graphite substrate during plasma spraying to produce molybdenum carbide (Mo_2_C) with good thermal stability [9,10,11]. The usage of a transition layer mainly aims at reducing large differences in physical properties between ceramic coating and base metal so as to loosen stress and avoid coating cracking, while the bonding force between coating material and base is enhanced [12].

Plasma spraying, a coating technique that combines physical vapor deposition (PVD), chemical vapor deposition (CVD), and slurry processes, is a reliable way to create thick ceramic coatings because of its benefits including a quick processing time and a larger range of spray and base materials [13]. For the purpose of overcoming structural flaws such as fracture, low adhesion strength, and porosity caused by thermal stress after plasma spraying, a laser can successfully modify the surface of the coating, so as to control the microstructure, reduce porosity, surface roughness, and improve durability [14]. Laser micromelting is primarily influenced by the laser sintering parameters, the laser beam’s optical characteristics, and the photothermal-mechanical characteristics of the substrate being treated. Thermal residual stress, microcracks, and a higher rate of material loss come from the melting and sputtering of materials as a result of the absorption of laser thermal energy [15].

A few researchers studied the impact of laser parameters on mechanical, thermal, and tribological properties by treating UHTCs with lasers. Wang et al. [16] studied MCrAlY coating sprayed by plasma with CO_2_ laser remelting. After laser remelting, the layered structure disappeared, density increased, porosity, inclusions, and other defects were basically eliminated, and the thermal corrosion performance of the coating was improved. In a vacuum environment, Lin et al. [17] studied pulsed laser ablation of bulk TaC substrates at high power and nanosecond pulses, for prospective tribological applications, the refractory particles’ bimodal minimum band gap seemed advantageous. A 400 W Nd:YAG laser was used by Pidani et al. [18] to study laser surface treatment of plasma-sprayed ceria-yttria stabilized zirconia (ZrO_2_). After heated corrosion, the volume fraction of m-ZrO_2_ in the sprayed and laser-glazed coatings grew to 86% and 39%, respectively. Additionally, Ref. [16] reports the ideal settings for pulsed Nd: YAG laser glazing of CYSZ TBCs. Utilizing a 2 kW CO_2_ laser, laser surface treatment of WC-Ni composite coatings was examined in order to determine the impact of laser power on wear rate and hardness. High laser energy levels decreased porosity and cracking while also diluting the coating. Lower laser energy levels could not connect to the substrate tightly enough [19].

The evaluation of scratch resistance of thermal spray ceramic coatings with laser surface treatment has always been the focus of research. The scratch resistance of laser remelted alumina (Al_2_O_3_) and chromia (Cr_2_O_3_) coatings used on carbon steel substrates was studied by Das et al. [20]. According to the authors, laser remelting boosts scratch wear resistance and failure load by 88% and 65%, respectively. However, during spraying, sputtering stratification occurred in the coating, and radial cracks were produced in the remelted coating. The effects of CO_2_ laser parameters on the remelting properties of substrates coated with (Al_2_O_3_) for austenitic steel were investigated by Marek et al. [21]. While the density of the structure was unaffected, an increase in laser energy resulted in the creation of pores. When compared to as-sprayed coatings, the critical load values for the laser remelted coatings were found to be greater.

Massive high-temperature structural ceramics [3,4,5,6,7,8] and other ceramics [16,17,18] have recently been the focus of intense study on laser surface treatment. There is limited research on laser remelting of plasma-sprayed structural ceramics [20,21]. However, the study of laser micromelting of plasma sprayed yttrium oxide coating on graphite substrate has not been reported. In order to avoid the macroscopic and microscopic defects of plasma sprayed UHTC coating, which are caused by instantaneous melting at high temperature, condensation, and tensile stress-induced phase transition process, the Gaussian light source fiber laser is used to micro-melt the coating, so as to prolong its service life, durability, and sustainability in the extreme environment. Therefore, the effects of laser power density, scanning speed, and defocusing distance on the bonding strength and scratch resistance of plasma sprayed Y_2_O_3_ coating form the focus of this work. Mo transition layer was deposited on graphite substrate by the Metco 3MB plasma spraying gun, and then Y_2_O_3_ coating was deposited. Microfusion experiments were carried out with a Fiber laser with a Gaussian light source. The microfusion performance indexes include the film base bonding performance and surface roughness, and the scratch performance was evaluated according to the critical load. An orthogonal test was used to evaluate the surface quality of Y_2_O_3_ coatings, and the mechanical properties such as film base bonding, surface porosity, and roughness were evaluated. X-ray diffraction (XRD) and scanning electron microscopy (SEM) were used to establish the relationship between the structure and properties of the coating.

## 2. Experimental-Procedure

### 2.1. Materials and Characterization

The atmospheric plasma spraying (APS) method was used to deposit Mo/Y_2_O_3_ coatings on the HDG substrate. Table 1 shows the plasma spraying parameters used to deposit Mo and Y_2_O_3_ coatings. To increase the adhesion between the Mo/Y_2_O_3_ coating and the graphite, the graphite surface was grit-blasted with Al_2_O_3_ particles (mesh size #60) at a pressure of 4.5 bar before plasma spraying. By utilizing a low-speed diamond wire cutter to slice the substrate along the cross section, coating thickness was measured. The cross-section and surface morphology of as-deposited coatings were observed under SEM with energy dispersive spectroscopy (EDS) (Carl Zeiss, F25, Oberkochen, Germany). Figure 1a shows the Y_2_O_3_ coating thickness of 180 μm obtained during plasma spraying and the elemental composition of the coating. Figure 1b is an EDS spectrum of the whole region. The scanning electron microscope image (500× magnification) was analyzed by ImageJ software (ImageJ, National Institutes of Health, New York, NY, USA), the porosity of the coating was calculated, and the image was adjusted to the appropriate threshold. The pore threshold was extracted from the background image by turning the pixels higher than the threshold into red, while the images lower than the threshold remained unchanged, so as to generate comparison in the region. The pixel area of the pore is measured relative to the total pixel area of the image. The percentage of porosity is calculated as the ratio of the pixel area of the pore to the total pixel area of the image [22]. The surface roughness of Y_2_O_3_ coating before and after micro-melting was measured by the JB-4C precision roughness tester (Shanghai Precision Instruments Co. Ltd., Shanghai, China). The microstructure of the coating was observed by an optical microscope and scanning electron microscope.

### 2.2. Laser Micromelting of Y_2_O_3_ Coated Graphite Substrate

In order to ensure the uniformity of the coating during micromelting, the lap rate of each line is set as 50% (the distance between each laser line is set as 0.5 mm), and the laser micromelting parameters (power density, scanning speed, and defocusing distance) are set as 50–100 W/mm^2^, 25–35 mm/s and 30–50 mm, respectively, used three factors and three levels of the orthogonal experiment design to repeat each experiment three times to ensure the repeatability of the process. The average values of coating porosity and surface roughness were calculated, and the experimental layout was given in Table 2.

### 2.3. Scratch Test and Tensile Strength Test of the Y_2_O_3_ Coating Sample

A ws-2005 automatic scratch instrument, equipped with a Rockwell diamond indenter with a tip radius of 200 μm and a tip angle of 120°, was used to evaluate the scratch behavior of the surface after spraying and laser ablation. The load was 0–30 N, the rate was 30 N/m, the scratch length was 4 mm, and the operation mode was dynamic load and unidirectional scratch. The tests were repeated three times in the same direction and average values were reported. The applied load acoustic emission (ae) signals and friction were recorded during the scratch test.

Tensile testing machine (PT-307, China) was used to test the tensile bonding strength of the coating. The maximum load is 10 kN, the tensile speed is set as 0.01 mm/min, the error of the test force indicator is <±0.05%, and the operation mode is dynamic loading and unidirectional drawing. The bonding strength between the coating and the substrate was reflected by the normal tensile stress borne by the coating per unit area. Equation (1) was used to repeat the experiment 7 times, where *P* is coating bonding strength, N/mm^2^ or MPa; *F* is the ultimate load of the coating under normal tensile force, N; *S* is the bonding area between coating and drawing rod, mm^2^:(1)$$P=\frac{F}{S}$$

## 3. Results and Discussion

### 3.1. Effect of Laser Parameters on Porosity of Y_2_O_3_ Coating

Laser micromelting is a complex process, which is affected by material properties and laser parameters such as laser power density, scanning speed, and defocus distance [23]. Therefore, the effects of laser power density (50–100 W/mm^2^), scanning speed (25–35 mm/s), and defocusing distance (30–50 mm) on the porosity and surface roughness were investigated. Experimental results of porosity and surface roughness are shown in Table 3. The influence of laser parameters on the porosity of Y_2_O_3_ coating is shown in Figure 2. The coating is a layered structure composed of numerous deformed particles interleaved and stacked together in a wavy manner. Inevitably, there are some pores and cavities between them, as shown in Figure 1b. With the increase of laser power density from 50 W/mm^2^ to 75 W/mm^2^, the average porosity decreases from 14.1% to 13.5%, mainly because the melting degree of particles on the coating surface increases with the increase of energy, which fills some pores and cavities. As the increase of laser energy leads to the increase of melting depth, the melting depth is proportional to the thermal diffusion depth, as shown in Equation (2) [24], which in turn increases the linear ablation rate:(2)$$L=I\ln\left(\frac{\varphi }{\varphi_{th}}\right)$$
where *L* is the melting depth of each laser, and *I* is the thermal diffusion depth of energy, which is determined by the thermal diffusion rate and laser duration. *φ* and *φ_th_* are laser integrating flux and ablation threshold integrating flux, respectively. The laser energy increases from 60 W to 80 W, and the porosity increases from 13.5% to 13.7%. This is because, under a certain laser energy, all the fusible particles on the surface of the coating are fused to fill the pores, and the thermal diffusion depth reaches the position of the base material, which leads to the thermal expansion of the base material and begins to spread the coating, resulting in the cracking of the coating, as shown in Figure 3.

When the scanning speed increases from 25 mm/s to 35 mm/s, the porosity increases from 13.3% to 14.2%, but the porosity is still less than 14.6% of the sprayed layer, indicating that the increase of scanning speed leads to the decrease of laser action time per unit area, resulting in the decrease of the thermal diffusion depth, and thus leads to the decrease of the melting degree of the coating surface. When defocus was increased from 30 mm to 50 mm, the porosity was reduced from 16.2% to 11.2%. On the one hand, the reduction of defocus leads to the formation of strong Gaussian heat flow, which leads to the ablation of part of the coating and the formation of ablation pits [25]. The formation of ablation pits can be confirmed by SEM images, as shown in Figure 3. On the other hand, with the increase of defocus distance, the spot diameter of the laser on the coating increases, and the fluence and the power density decreases, which reduces the thermal diffusion depth on the substrate and reduces the thermal expansion of graphite substrate on the coating. According to [26], a small amount of defocus within a certain laser power density range will lead to the discharge, rather than the vaporization of the material. This causes an increase in the porosity and surface roughness of the coating.

### 3.2. Effect of Laser Parameters on Surface Roughness of Y_2_O_3_ Coating

The influence of laser micromelting parameters on the surface roughness of Y_2_O_3_ coating is shown in Figure 4. As the laser power density increases from 50 W/mm^2^ to 75 W/mm^2^, the surface roughness of Y_2_O_3_ coating decreases from 6.568 μm to 6.464 μm, which may be due to the slow cooling rate of the molten pool on the top surface of Y_2_O_3_ at a higher pulse energy [27]. The surface roughness of Y_2_O_3_ coating increases from 6.464 μm to 6.474 μm when the laser power density increases from 75 W/mm^2^ to 100 W/mm^2^. Based on the SEMs, the pits are not small enough to count as 10 nm begin to appear with the increase of laser energy. When the laser scanning speed increases from 25 mm/s to 35 mm/s, the roughness value decreases from 6.521 μm to 6.363 μm, and the interaction time between the surface and the laser beam decreases, the input energy per unit area also decreases, and the rise of surface temperature is limited, leading to the cooling rate becoming faster. Therefore, lower surface roughness is observed [28]. However, as the scanning speed increases from 30 mm/s to 35 mm/s, the roughness value increases from 6.363 μm to 6.621 μm because there is not enough time to melt the surface of the coating [29]. The surface roughness decreases from 6.934 μm to 6.119 μm with the defocus increasing from 30 mm to 50 mm. This is mainly because the increase of defocus distance leads to more uniform heating of the coating surface during laser micromelting, and the surface powder particles and protrusions are fused, leading to the improvement of surface quality.

### 3.3. The Relationship between the Structure and Properties of Spray Surface and Micro-Fused Surface

In the plasma spraying process, Y_2_O_3_ coating reacts with the Mo transition layer at a high temperature to form part of Y_2_MoO_6_ and Y_6_MoO_12_, thus improving the bonding properties of the coating. The XRD patterns of Y_2_O_3_ coated graphite (T0) and laser micromelted Y_2_O_3_ coated graphite (T5) and (T9) are shown in Figure 5. The coating phase did not change. With the increase of laser power, the peak values of (T0), (T5), and (T9) increase overall, and the strongest peak diffraction Angle θ is 14.5°, 14.54°, and 14.66°, respectively. The grain size was calculated as 333 nm, 304 nm, and 276 nm by the Scherrer formula, which indicates that the degree of crystallization of yttrium oxide on the coating surface increases, and the crystal plane spacing decreases. It shows that the mechanical properties of the coating can be effectively improved and the scratch resistance can be better due to the rapid cooling of the coating and some degree of fine grain strengthening. In the process of laser micromelting, it is necessary to control the depth of thermal influence by adjusting the laser power density, scanning speed, and defocusing distance, it is necessary to ensure that the Y_2_O_3_ particles on the surface of the coating melt, and to let the graphite substrate not expand for overheating as well as the Mo element not spread to the surface of the coating. The surface EDS diagram of (T5) is shown in Figure 6a. No C and Mo elements are precipitated on the surface of Y_2_O_3_ coating. It can be clearly seen from the EDS diagram of in Figure 6b that Y_2_O_3_ coating, Mo coating, and graphite substrate have partial mutual diffusion, which is conducive in improving the bonding performance of the coating. Figure 6c,d are the SEM images of section (T6) and (T8), respectively. (T6) shows partial transverse microcracks, and (T8) shows a large number of longitudinal cracks while the coating shows signs of spalling. This indicates that, at the same power density, when the defocusing distance decreases from 50 mm to 30 mm, the depth of thermal influence in the laser micromelting process will be increased, resulting in transverse cracks in the internal expansion of the coating. However, increasing the power density and decreasing the defocus distance will cause the heat-affected area to reach the graphite substrate, and the thermal expansion of the graphite substrate will lead to longitudinal cracks and begin to peel off the coating. After laser micromelting, as shown in Figure 7a, is the SEM figure of Y_2_O_3_ coated graphite (T0). Figure 7b–d are respectively after micromelting (T3), (T5), (T7) with power density 50 W/mm^2^, 75 W/mm^2^, and 100 W/mm^2^ and defocusing distance 50 mm. It can be clearly seen that the large particles on the surface of Y_2_O_3_ coating melt gradually with the increase of power density. When the power density reaches 100 W/mm^2^, the particles on the surface of the coating basically disappear, while a large number of cracks appear.

### 3.4. Comprehensive Evaluation of Critical Load and Tensile Strength Test for a Scratch Test of Laser Micromelted Y_2_O_3_ Coating

In order to evaluate the critical load, progressive load scratch tests were carried out on sprayed and laser-ablated Y_2_O_3_ coated graphite substrates. Critical load is defined as the minimum load at which the coating begins to fail and continues to fail at higher loads [30]. At the same time, the tensile strength test of the coating was carried out to comprehensively evaluate the properties of the coating. Figure 8 shows the acoustic emission signals, tangential friction curves, and corresponding scratch diagrams of spray state (T0) and laser ablated Y_2_O_3_ coating (T0)–(T9), respectively. (T0), (T1), (T2), (T3), (T7), (T8), and (T9) can obviously see the wedge spalling of the coating, which may be due to the high tensile stress generated during the head test, while the compressive stress caused by friction at the front of the head is relatively low [31]. Combined with Figure 6d and Figure 7a,c,d, the reasons for this phenomenon can be divided into two types: one is that the coating surface contains Y_2_O_3_ powder particle clusters, microcracks, and high surface pores generated in the plasma spraying process or not fused after laser micro-melting; the other is that some transverse cracks, longitudinal cracks and spalling caused by thermal expansion appear in the coating after laser micro-melting. Thus, the bearing capacity of Y_2_O_3_ coating is reduced, and the critical load is lower. A comprehensive evaluation of (T4), (T5), and (T6) was carried out based on the bar chart of the critical load and tensile strength of the coating in Figure 9. It was found that no obvious peeling phenomenon of the coating was found after the scratch test, and the critical load of (T5) and (T6) was significantly increased compared with that of spray coating (T0). However, in the tensile strength test, the tensile strength of (T5) 5.68 MPa is 2.03 MPa higher than that of (T6) 3.65 MPa, which is because of transverse micro-cracks in the middle of (T6) coating. As shown in Figure 6c, such transverse cracks are more likely to cause coating failure under tensile stress. Therefore, after comprehensive evaluation, when the laser power density is 75 W/mm^2^, the scanning speed is 30 mm/s, and the defocusing distance is 50 mm, the laser micro-melting of the sprayed Y_2_O_3_ coating can greatly improve the compressive and tensile properties of the coating, and ensure that the inner structure of the coating is tight and the element diffusion degree of the substrate and transition layer is small.

## 4. Conclusions

The effects of laser power density (50–100 W/mm^2^), scanning speed (25–35 mm/s), and defocusing distance (30–50 mm) on the surface morphology and scratch resistance of Y_2_O_3_ coated graphite substrate were studied by using continuous fiber laser technology. Under the conditions of laser power density of 75 W/mm^2^, scanning speed of 30 mm/s, and defocusing distance of 50 mm, the minimum surface porosity of 10.2% and surface roughness of 5.826 μm can be obtained simultaneously. The surface with the smallest surface porosity and roughness has a higher critical load of 18.05 N and the highest tensile strength of 5.68 MPa. At the same time, the inner structure of the coating is close without any cracks, that is, it has the largest scratch resistance. The fast cooling characteristics of laser micro-melting make the coating surface form fine grain strengthening and improve the mechanical properties, and the failure is mainly due to the spallation of some particles at the end of the scratch.

## Figures and Tables

**Figure 1 materials-15-05443-f001:**
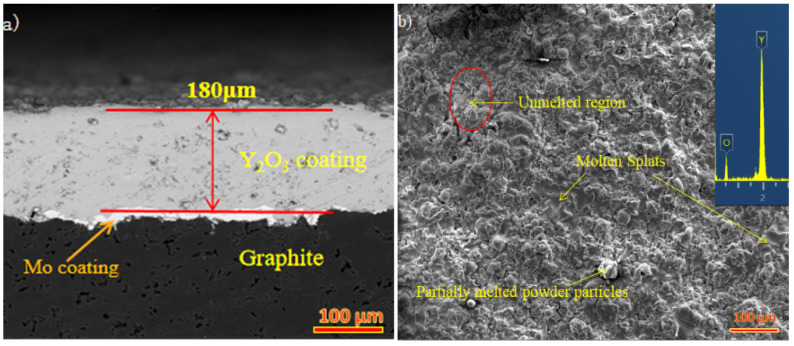
(**a**) Sectional view of coating; (**b**) microstructure of coating with EDS over the entire area.

**Figure 2 materials-15-05443-f002:**
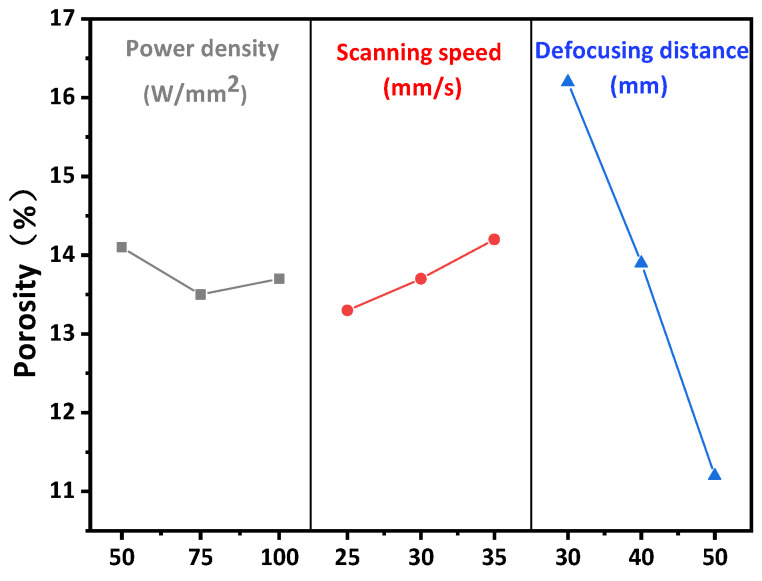
Effect of laser micromelting parameters on porosity of Y_2_O_3_ coating.

**Figure 3 materials-15-05443-f003:**
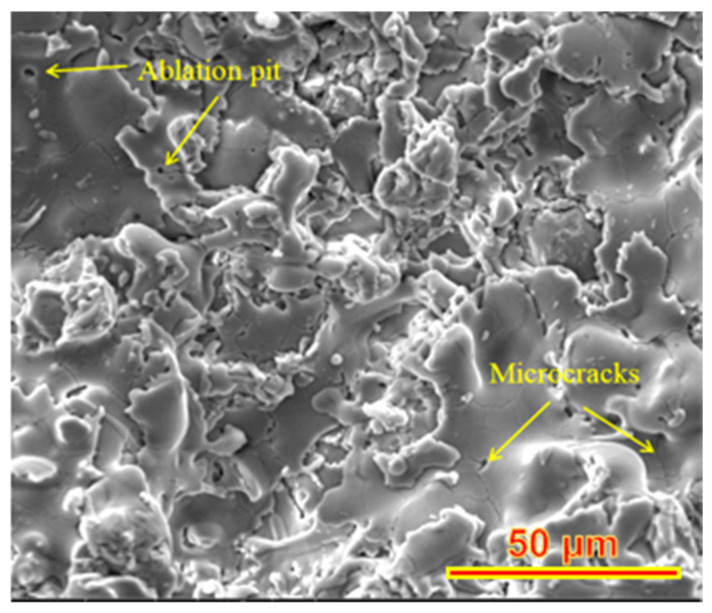
Microstructure of Y_2_O_3_ coating after laser micromelting.

**Figure 4 materials-15-05443-f004:**
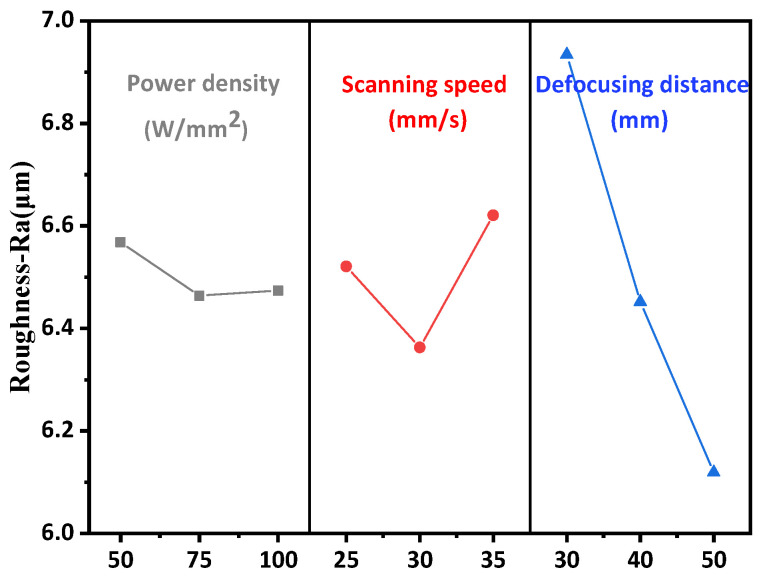
Effect of laser micromelting parameters on surface roughness of Y_2_O_3_ coatings.

**Figure 5 materials-15-05443-f005:**
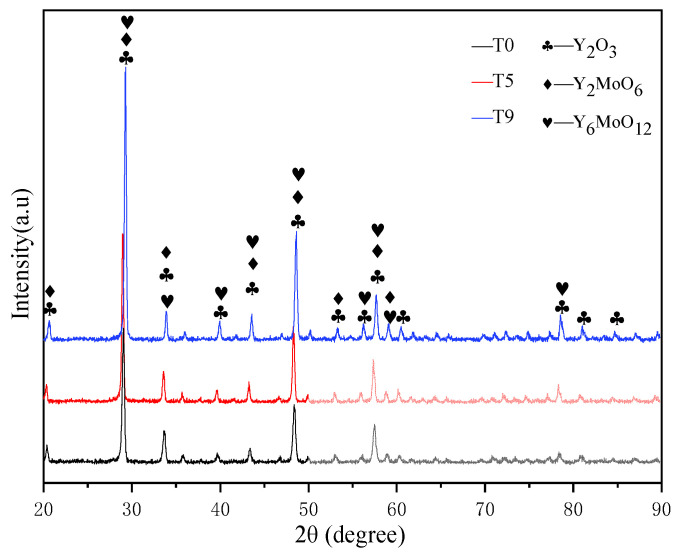
XRD patterns of Y_2_O_3_ coated graphite (T0) and laser micromelted Y_2_O_3_ coated graphite (T5) and (T9).

**Figure 6 materials-15-05443-f006:**
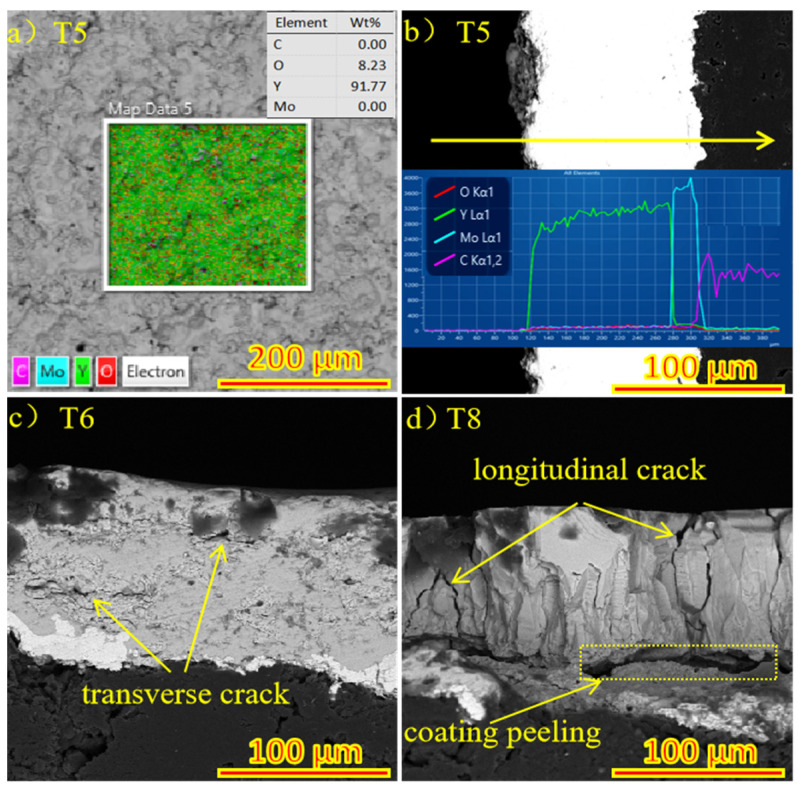
(**a**) The surface EDS of (T5); (**b**) The section EDS of (T5); (**c**) The section SEM of (T6); (**d**) The section SEM of (T8).

**Figure 7 materials-15-05443-f007:**
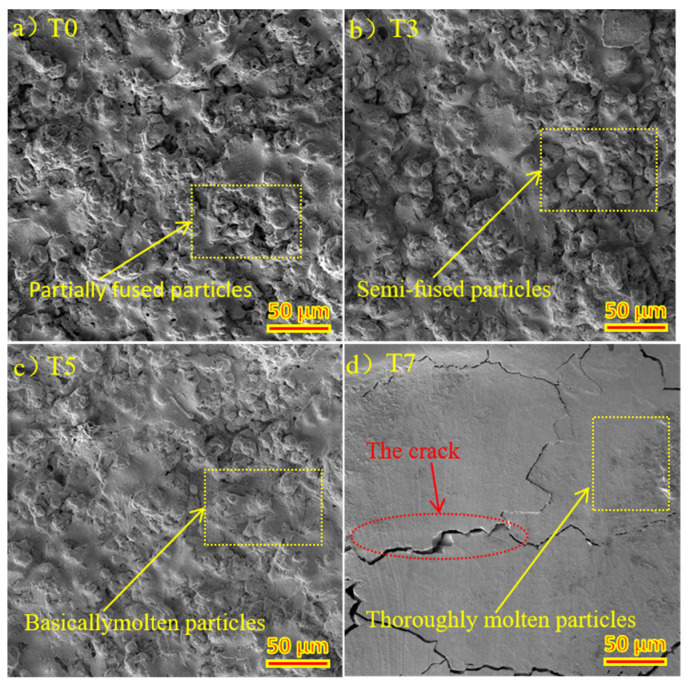
(**a**) (T0) coating surface microstructure; (**b**) (T3) coating surface microstructure; (**c**) (T5) coating surface microstructure; (**d**) (T8) coating surface microstructure.

**Figure 8 materials-15-05443-f008:**
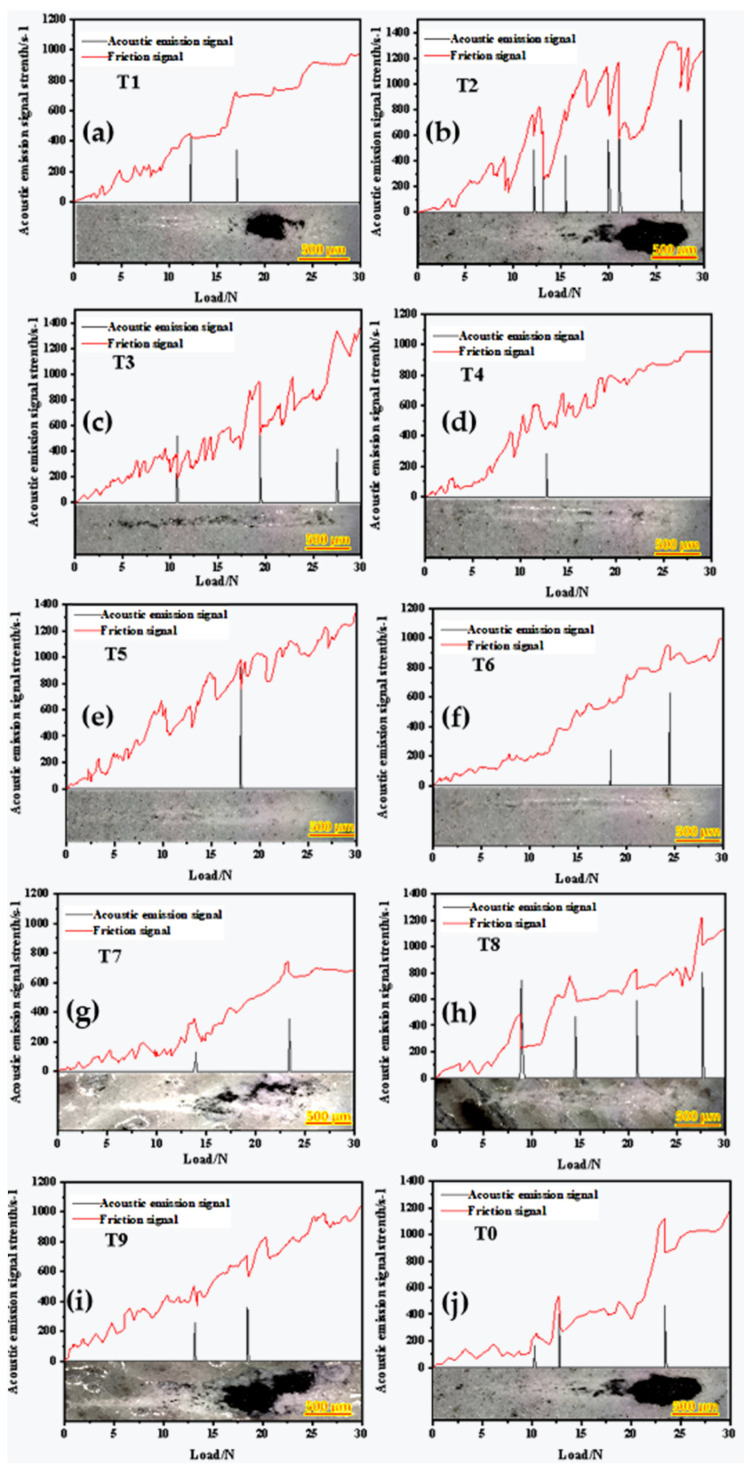
(**a**) (T1) scratch behavior; (**b**) (T2) scratch behavior; (**c**) (T3) scratch behavior; (**d**) (T4) scratch behavior; (**e**) (T5) scratch behavior; (**f**) (T6) scratch behavior; (**g**) (T7) scratch behavior; (**h**) (T8) scratch behavior; (**i**) (T9) scratch behavior; (**j**) (T0) scratch behavior.

**Figure 9 materials-15-05443-f009:**
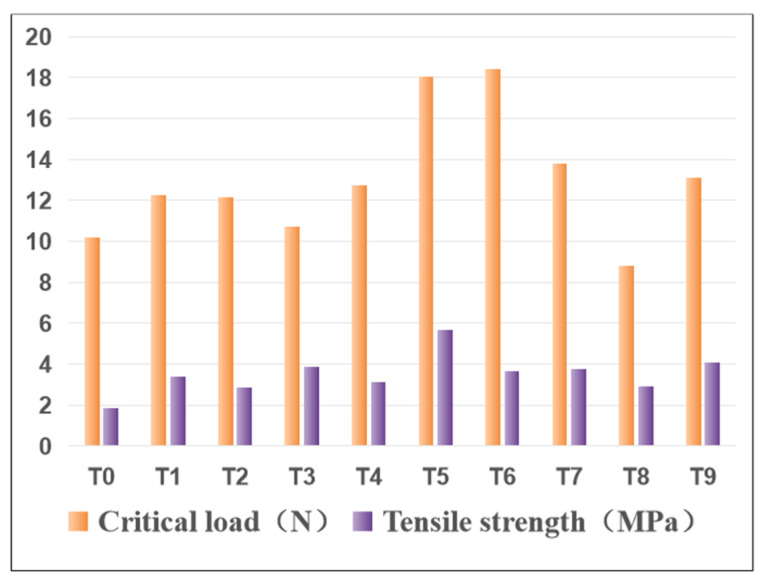
Critical load and tensile strength of coating.

**Table 1 materials-15-05443-t001:** Plasma spraying parameters of Mo transition layer and Y_2_O_3_ coating.

Plasma Spray Parameters	Molybdenum Coating	Yttria Coating
Arc current (A)	450	750
Arc voltage (V)	50	40
Primary argon gas flow (LPM)	40	45
Secondary H_2_ gas flow (LPM)	4	2
Standoff distance (mm)	100	100

**Table 2 materials-15-05443-t002:** Experimental layout of laser micromelting parameters.

Experiment No.	Power Density (W/mm^2^)	Scanning Speed (mm/s)	Defocusing Distance (mm)	Mean Porosity (%)	Mean Surface Roughness -Ra (μm)
T1	50	25	30	15.3	6.903
T2	50	30	40	14.2	6.451
T3	50	35	50	12.7	6.35
T4	75	25	40	13.9	6.480
T5	75	30	50	10.2	5.826
T6	75	35	30	16.5	7.087
T7	100	25	50	10.7	6.182
T8	100	30	30	16.8	6.813
T9	100	35	40	13.5	6.426
T0	-	-	-	14.6	6.98

**Table 3 materials-15-05443-t003:** T value analysis of laser micromelting reaction.

T-Value	Porosity (%)			Surface Roughness -Ra (μm)		
	Power Density	Scanning Speed	Defocusing Distance	Power Density	Scanning Speed	Defocusing Distance
K1	42.2	39.9	48.6	19.704	19.565	20.803
K2	40.6	41.2	41.6	19.393	19.09	19.357
K3	41	42.7	33.6	19.421	19.863	18.358
t1	14.1	13.3	16.2	6.568	6.521	6.934
t2	13.5	13.7	13.9	6.464	6.363	6.452
t3	13.7	14.2	11.2	6.474	6.621	6.119
R	0.6	0.9	5	0.104	0.258	0.815

## Data Availability

The data presented in this study are available on request from the corresponding author.

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
