# Peer review of "Tuning the Surface Characteristics and Mechanical Properties of Y2O3 Coatings on a Graphene Matrix via Laser Micro Melting"

_materials, 2022, doi:10.3390/ma15155443_

Round 1

Reviewer 1 Report

Minor grammar/syntax correction throughout the text (see attached pdf). I also suggest trying to rewrite most of your explanations for deconvolution/simplicity.

Need to explain why a more robust technique for porosity determination has not been applied (e.g. BET/BJH porosimetry), and you had to rely on graphical analysis of SEMs.

XRD analysis very poor. I suggest giving Table with all associated metrics (2theta, d-distance, cryst. size, Miller indices etc.) and to include Bragg lines in the graph.

Reviewer 2 Report

the manuscript materials-1851939 whose title is "Tuning the surface characteristics and mechanical properties of Y2O3 coatings on the graphene matrix via laser micro melting" by Hao Liu and co-workers fits well the scopes of the MDPI' journal Materials since concerns the study of laser microstructural surface treatments in order to overcome structural defects in thick ultra-high temperature ceramic coatings of yttrium oxide obtained by plasma techniques.
The authors start with a well-structured introduction in which the topic background and the reason for the choice of the proposed laser technique with respect to the final applications of the ceramic materials are explained properly. The manuscript describes the experimentation of laser parameters to induce changes in the surface's microstructure and, therefore, in the morphological and mechanical properties of coatings of Y2O3 sprayed on graphite substrates by plasma spray technique (which is one of the useful method to prepare thick ceramic coatings). Furthermore, since the main drawback related to the plasma spray technique is the possibility to induce fracture, excessive porosity and therefore low adhesion strength, the authors proposed to overcome these probable drawbacks by the laser micromelting strategy. Not many researchers studied the implications of these laser treatments, especially micromelting, on ultra-high temperature ceramics therefore the manuscript here proposed can contribute to fill this gap in this research topic.

In general, the manuscript is very well written and easy to understand. The topic discussed is sufficiently new and interesting to the research community working on surface changes by micromeltig induced by laser ablation of ultra-high temperature ceramic coatings, such as yttrium oxide, useful for specific applications in which the resistance of coating at ultra-high temperature is highly needed.
In my opinion, the details about experimental conditions are adequate and clearly explained affecting positively the manuscript quality. Moreover, the conclusions together with the discussion section are well developed and the explanation of the relationship between the laser parameters (power density, scanning speed and defocusing amount) and the properties of the resulting micromelted yttrium oxide coating are adequately reported by taking advantages from SEM, XRD, EDS techniques as well by the analysis of critical load, tensile strength  and scratch behavior of resulting treated coatings.
Since the here presented work could be very useful for scientists working on these topics, from my point of view, the manuscript could be published after the minor revisions already reported in the form for authors.

Page 2, line 75: please, check the phrase here reported since it has incorrect punctuation.
Page 3, section 2.1: please, adequate how to cite figures. In the rest of the manuscript, you always used the style: "Figure x" instead in this section you referred to figure 1 as "FIG.1".
Page 4, line 147: the authors reported the phrase in which they mentioned the "ae signal" but this acronym has not been defined before. Please, provide the definition of this acronym.
Page 8, Figure 6: please, check the caption of this figure.
